# Experimental Measurements of the Length of the Human Colon: A Systematic Review and Meta-Analysis

**DOI:** 10.3390/diagnostics14192190

**Published:** 2024-09-30

**Authors:** Faiz Alqarni, Tejal Akbar, Hala Fatani, Soma Kumasaka, Caroline L. Hoad, Robin C. Spiller, Moira A. Taylor, Luca Marciani

**Affiliations:** 1Translational Medical Sciences, Nottingham Digestive Diseases Centre, School of Medicine, University of Nottingham, Nottingham NG7 2UH, UK; faiz.alqarni@nottingham.ac.uk (F.A.); tejal.akbar@nottingham.ac.uk (T.A.); stxhf@nottingham.ac.uk (H.F.); robin.spiller@nottingham.ac.uk (R.C.S.); 2National Institute for Health and Care Research (NIHR), Nottingham Biomedical Research Centre, Nottingham University Hospitals NHS Trust, University of Nottingham, Nottingham NG7 2UH, UK; caroline.l.hoad@nottingham.ac.uk (C.L.H.); moira.taylor@nottingham.ac.uk (M.A.T.); 3King Saud Medical City, Ministry of Health, Riyadh 11196, Saudi Arabia; 4Department of Diagnostic Radiology and Nuclear Medicine, Gunma University Graduate School of Medicine, Maebashi 371-8511, Japan; soma.kumasaka@nottingham.ac.uk; 5Radiological Sciences, School of Medicine, University of Nottingham, Nottingham NG7 2UH, UK; 6Sir Peter Mansfield Imaging Centre, School of Physics and Astronomy, University of Nottingham, Nottingham NG7 2RD, UK; 7The David Greenfield Human Physiology Unit, School of Life Sciences, University of Nottingham, Nottingham NG7 2UH, UK

**Keywords:** colon length, large intestine length, large bowel morphology, colon measurements, colon anatomy

## Abstract

Purpose: Knowledge of the length of the colon is relevant to understanding physiological and pathological function. It also has implications for diagnostic and clinical interventions, as well as for the design of delayed-release drug formulations and drug disposition modeling. Methods: Over the years, a range of different experimental methods have been employed to assess colon length. These methods vary from direct measurements on cadavers and during intraoperative procedures to measurements obtained from endoscopic and medical imaging techniques. However, no systematic review or meta-analysis of these findings has yet been carried out. In this systematic review, we identified 31 published experimental studies that measured the length of the human colon and/or its segments. Results: We synthesized the available data, comprising colon length measurements from 5741 adults and 337 children and young people, in a meta-analysis. The data contribute to our understanding of colon morphology and may have implications for clinical practice, particularly for colonoscopy and preoperative planning of surgical resections. Additionally, this review provides potential insights into anatomical correlates of functional diseases, such as constipation. Conclusions: This review highlights that non-invasive, non-destructive diagnostic imaging techniques, such as magnetic resonance imaging (MRI), can provide more physiologically relevant measurements of colon length. However, there is a need for more standardized measurement protocols and for additional pediatric data.

## 1. Introduction

The colon is one of the main organs of the human gastrointestinal tract, performing crucial functions such as the absorption of water and electrolytes, transport of chyme, and the formation and elimination of feces. The colon begins at the cecum, where the small intestine meets the large intestine, and ends at the rectum. The ascending colon travels upward on the right side and ends at the hepatic flexure. The transverse colon crosses the abdomen from right to left and ends at the splenic flexure. The descending colon then travels from the splenic flexure downward on the left side and transitions into the sigmoid colon at the level of the pelvic brim, which curves into the pelvic region before connecting to the rectum [1]. The clinical outcomes of procedures such as colonoscopy and surgery can be affected by variability in colon anatomy and dimensions [2,3]. Knowledge of colon length can help when planning surgical procedures and in understanding conditions like constipation. For example, colonic elongation has been related to slow transit constipation and functional fecal retention [4,5]. The length of the colon has also been correlated with the presence of diverticula [6]. Colon length measurements are also needed to interpret high-end colon motility measurements [7] and to inform the design of delayed-release drug dosage forms and drug disposition modeling.

Despite the importance of this organ, knowledge of its morphology and particularly of its length is still incomplete. Over the years, a range of different experimental methods have been employed to assess colon length. These vary from measurements on cadavers and during intraoperative procedures to endoscopic and medical imaging techniques, but no systematic review of these findings has yet been carried out.

Therefore, this review and meta-analysis aimed to systematically identify, evaluate, and synthesize studies that measured the length of the colon and/or its distinct anatomical segments using experimental methodologies.

## 2. Materials Methods

This review followed the guidelines outlined in the Preferred Reporting Items for Systematic Reviews and Meta-analysis of Diagnostic Test Accuracy Studies (PRISMA-DTA) statement [8]. The protocol was developed in advance of conducting the work and it was registered with the International Prospective Register of Systematic Reviews (PROSPERO, registration number CRD42023465879).

### 2.1. Search Strategy

A comprehensive electronic search was carried out across three databases: Web of Science, MEDLINE (Ovid), and EMBASE (Ovid). Date restrictions were not applied. The initial search was conducted in September 2023, and it was last updated in May 2024. Advanced literature search techniques were employed using “AND” and “OR” operators to refine or broaden the search scope. Boolean search strategies were utilized to locate primary sources, and relevant journal paper references were examined. The search terms were expanded beyond colon length, incorporating synonyms and varying search words such as (colon anatomy or colon morphology) to explore a broader spectrum of hits. Reference chaining and manual searches on alternative engines such as Google and Google Scholar were carried out to identify potential hits not found in the primary databases. All references retrieved from electronic databases were imported into EndNote reference management software version 20.5 (Clarivate, London, UK).

The search strategy and keywords used in the MEDLINE and Embase (OVID) database were as follows:1—((colon or colorect* or colonic*) adj1 (length* or measur* or size* or anatomy*)).ti,ab.2—(“large intestine” adj1 (length* or measur* or size* or anatomy*)).ti,ab.3—(“large bowel” adj1 (length* or measur* or size* or anatomy*)).ti,ab.4—(rat or rats or mouse or mice or swine or porcine or murine or sheep or lambs or pigs or piglets or rabbit or rabbits or cat or cats or dog or dogs or cattle or bovine or monkeys or trout or marmoset $1).ti. and animal experiment/.5—animal experiment/ not (human experiment/ or human/).6—4 or 5.7—1 or 2 or 38—7 not 6.

### 2.2. Eligibility Criteria

Selection of studies was carried out following predefined criteria.

Eligible studies for the review included randomized controlled trials, cross-sectional studies, case–control studies, cohort (longitudinal) studies, observational studies (of all types), cohort studies, and non-randomized controlled trials that measured the length of the colon and/or its distinct anatomical segments using various experimental methodologies. All methodologies were eligible, including computed tomography (CT), magnetic resonance imaging (MRI), X-ray radiography, surgical approaches, and cadaveric investigations. Only studies involving human participants (including adults, children, males, and females, healthy and patients) were considered for inclusion. The primary outcome of interest was colon length, either the total length of the colon and/or the lengths of specific anatomical segments (namely, the cecum colon, ascending colon, transverse colon, descending colon, sigmoid colon, and rectum). The review also aimed to appraise the various techniques and tools used to measure colon length.

Exclusion criteria were studies not published in English language, case reports, review articles (including meta-analyses), editorials, replies, opinion pieces, conference abstracts, meeting abstracts, books, or book chapters. All non-human studies were excluded as well.

### 2.3. Study Selection and Data Extraction

All references retrieved from the three databases underwent a systematic review process. Firstly, two reviewers (F.A. and T.A.) removed duplicate papers across the databases. They then independently screened and filtered titles and abstracts based on the above inclusion and exclusion criteria. Full manuscripts of the identified studies were then sourced. They then assessed the eligibility of the studies on the full manuscripts, providing justifications for their inclusion and exclusion decisions. For any disagreement, the initial two reviewers discussed the matter based on the criteria and if agreement could not be reached, the discussion was escalated for moderation by a third reviewer (L.M.) until consensus was reached. A PRISMA flowchart was constructed to capture the selection process of both included and excluded studies (Figure 1).

Following this, two reviewers (F.A. and H.F.) carried out data extraction from the included papers and designed a summary table containing author details, publication date, location, study type, population characteristics, age distribution, sex ratio, presence or absence of a comparison group, and colon assessment techniques. An additional reviewer (L.M.) was enlisted to assist with data extraction and to moderate cases of disagreement through discussion.

### 2.4. Quality Assessment of Included Studies

Quality and bias assessments of the included studies were conducted by the same two reviewers (F.A. and H.F.) using the Newcastle–Ottawa Quality Assessment Scale (NOS), adjusted for cross-sectional studies [9]. The scale comprises three categories: selection, comparability, and outcome, with a maximum total score of 10 points for all categories. Selection criteria include representativeness, sample size, nonexposed cohort, and measurement tool, with a maximum score of 5 points. Comparability assesses the measurement of confounding factors, with a maximum score of 2 points. Outcome evaluates outcome assessment and statistical tests, with a maximum score of 3 points. Each category has different scoring scales. Reviewers assign scores based on these criteria, with higher scores indicating higher research methodology quality.

The NOS criteria were modified based on the review requirements [9]. The bias assessment criteria included participant selection (type, age, sex), a sample size of more than twenty participants, clear descriptions of measurement tools and techniques, comparability, type of assessment, and statistical tests used, providing a comprehensive framework for bias assessment in the analyzed cross-sectional studies. Quality and risk of bias for the studies included in the review were independently evaluated by two reviewers (F.A. and H.F.). These studies varied in study type, with the majority being retrospective. Assessment scoring ranged from 0 to 10 points, with 0–4 points considered low quality, 5–6 satisfactory, 7–8 good, and 9–10 high quality.

### 2.5. Meta-Analysis Methods

All mean values of the length of the colon and/or of its segments were extracted from the papers reviewed in order to combine them and determine overall mean values and variability. The pediatric values varied considerably with age; hence, the meta-analysis was split into two parts. All the adult colon and colon segment lengths were grouped together. Secondly, the pediatric total colon length values were plotted against age; there were only a few data available for individual colon segment lengths and therefore these were not pooled into an overall analysis.

## 3. Results

### 3.1. Search Results

Twenty-two papers satisfying the inclusion/exclusion criteria were identified and included for data extraction. Eight additional papers were identified through cross-referencing plus one from the grey literature, bringing the total to thirty-one included papers. Overall, this body of work comprised colon length measurements carried out on 5741 adults and 337 children and young people. The literature search hits, identification, and screening process are summarized as a PRISMA diagram in Figure 1.

### 3.2. Quality Assessment of Included Studies

Figure 2 shows the NOS assessment result. Varying levels of bias were identified as follows: seven studies were rated as high quality, eighteen as good, four as satisfactory, and two as low, particularly in terms of selection and comparability of research groups, and determination of exposure or outcome. Common issues included non-representative samples, insufficient control for confounding variables, and high non-response rates, contributing to higher bias levels. While some studies exhibited strong methodologies and comparability, others lacked clarity.

### 3.3. Demographics of the Included Studies

Table 1 provides an overview of the demographics in the 31 studies included. These studies examined colon length using various methods and techniques, with some assessing the entire colon while others focused on specific segments. Among the included studies, eleven were conducted in the UK [2,10,11,12,13,14,15,16,17,18,19] with three of them performed by the same research group [12,13,14]. Five studies originated from the USA [20,21,22,23,24], four from Japan [25,26,27,28], two conducted by the same research group [25,28], two from India [29,30], and one each in Australia [31], Ethiopia [32], France [3], Italy [6], New Zealand [33], South Africa [34], Denmark [7], Canada [35], and Nigeria [36].

### 3.4. Colon Length Assessment Methods

Table 2 presents the data on colon length extracted from each study, categorizing separately measurements carried out on total colon length and/or separate anatomical segments, which reflects the respective study designs and purposes. The studies were classified according to the techniques employed in assessing colon length. The measurement values listed in Table 2 are all expressed in centimeters.

#### 3.4.1. Cadaver Assessment Studies

Nine studies carried out colon measurements on cadavers (Figure 3) utilizing various measurement tools, including surgical tape or a ruler [3,10,11,18,20,29,30,31,36]. Some of these studies included comparisons between cadavers and living subjects [29,31,36]. Some of these investigations evaluated both total colon length and length of segments, while others specifically focused on individual segments, such as the sigmoid colon [29,30,36]. A strong positive relationship between body weight and colon length was also suggested [3].

#### 3.4.2. Intraoperative Assessment

Five studies [13,29,32,35,36] assessed colon length in patients undergoing surgical procedures such as laparotomy or for conditions like sigmoid volvulus. Measurements were carried out using tape or suture thread. Some investigations involved comparisons between living subjects and cadavers [36], while others compared patients with and without a history of sigmoid volvulus [32]. This study found that a sigmoid colon that is elongated with a long and wide mesentery, but maintains a consistent base, is strongly associated with an increased risk of sigmoid volvulus [32].

#### 3.4.3. Colonoscopy Assessment Studies

Five studies [15,16,22,24,26] employed colonoscopy to evaluate colon length. Colonoscopy involves the insertion of a camera catheter from the ano-rectum through the colon for diagnostic purposes and can also enable colon length measurements. These studies utilized various colonoscopy techniques, incorporating aids such as computer-aided measurements or magnetic endoscopic imaging (MEI). MEI utilizes small radio transmitter coils placed along the catheter that allow three-dimensional tracking of the catheter as it moves around the colon [15,16]. Some of these investigations also conducted simultaneous comparisons with alternative techniques, such as computed tomography colonography (CTC) [22,24,26], to investigate possible differences. There was a significant difference in overall colonic length between CT colonography and conventional colonoscopy. The number of acute angle flexures and the degree of tortuosity detected with CT colonography were higher than expected [24].

#### 3.4.4. Radiological Barium Enema Assessment Studies

Six studies [12,14,25,28,31,34] employed the radiological barium enema technique, which exploits X-ray imaging of the colon and involves prior administration of luminal contrast mediums containing barium to provide X-ray contrast (Figure 4). Measurements are then carried out on the films using tools such as opisometer wheels, which is an instrument for measuring the lengths of arbitrary curved lines, or ruler measures to measure the colon either in its entirety and/or by segment. Some studies included comparisons of this technique with alternative methods, such as cadaveric measurements [31], while others evaluated only the effectiveness of the barium enema imaging technique itself. The comparison of barium enema with cadaver studies for colon length measurement indicated that the barium enema method was a reliable tool compared to the cadaveric measurement [31].

#### 3.4.5. Computed Tomography Colonography Assessment

Ten studies [6,17,19,21,22,23,24,26,27,33] carried out colon length measurements using CT colonography (CTC). CT utilizes X-ray beams rotating around the patient, which allows three-dimensional imaging of the abdomen. The measurement techniques utilized across these studies are largely uniform, employing automated centerline measurements to generate a 3D map (Figure 5). Additionally, some of these investigations compared the efficacy of CTC methods with traditional colonoscopy techniques [22,24,26]. CT colonography (CTC) is a diagnostic procedure designed to detect colonic adenomatous lesions and is less invasive than colonoscopy and it allows examining colonic morphology at the same time [26].

#### 3.4.6. Magnetic Resonance Imaging and 3D-Transit Assessment

Two studies [2,7] explored colon length using MRI techniques (Figure 6). MRI uses radio waves to transmit and receive a signal from the water inside the body; the signal is localized spatially, and three-dimensional images of the abdomen can be reconstructed. One of the studies compared MRI findings with another remote tracking system called 3D-Transit, which utilizes electromagnetic tracking capsules [7]. The study showed that the 3D-Transit method reliably estimated colorectal length with a consistent level of accuracy when compared to MRI-derived measurements, showing acceptable variation. Moreover, it demonstrated reliability across recordings conducted on two consecutive days [7]. The second study utilized image processing visualization software to compare colon characteristics between healthy children and those with functional constipation [2].

### 3.5. Meta-Analysis

All the available mean values of the length of the adult colon and/or its anatomical segments, for all techniques, are plotted in Figure 7 and the descriptive numerical values are reported in Table 3. The data show the variability in the values reported, which have coefficients of variation between 19% and 39%, with the sigmoid colon being the most variable length. Meta-analysis of the data showed that the overall mean of the mean lengths of the adult colon was 148.3 cm with a standard deviation of 27.1 cm.

Fewer values were available for the mean length of the pediatric colon, with the values mostly reflecting the total colon. This is plotted against the age of the children in Figure 8, showing a semi-log increase in total colon length with age (R^2^ = 0.4012, *p* < 0.0333).

## 4. Discussion

The data from the 31 publications identified and reviewed here provided a new comprehensive picture of current knowledge of the length of the human colon and of its anatomical segments.

The body of data reviewed originates from many studies using very different techniques and spanning over a century. They include different populations, different ethnicities, and potential functional and neurological confounds. This needs to be taken into account when reflecting on the data and the variability between the measurements, particularly in the meta-analysis. Direct measurements were performed during surgical procedures and on cadavers. A study comparing living subjects and cadavers highlighted marked differences in sigmoid colon lengths, which were 64% longer in living subjects compared with cadavers [29]. However, another study found no marked differences between living and cadaveric sigmoid colon lengths [36]. Comparison between radiographic studies and postmortem studies also showed differences, but with postmortem lengths being higher than for radiographic studies [31]. Some of these differences may be due to postmortem/positional changes compared with the living organ in situ. Radiographic studies are also limited to two-dimensional projections of the organs, which do not reflect antero-posterior dimensions, which can introduce larger errors for the distal part of the colon. By contrast, cross-sectional medical imaging techniques can provide a better three-dimensional appraisal of the colon. Colonoscopy measures may depend on the maneuvering technique used to pass the camera catheter through the colon, which could yield underestimated measures of length, and in some reports, CTC was shown to overestimate colon length compared with colonoscopy [22,24]. The use of barium enemas and of CT luminal contrast are not physiological and may also affect the morphology before measurement. Drugs used for colonoscopy and anesthetics used for surgical procedures may also increase colon length. MRI in the undisturbed colon was a more recent addition to the above techniques [2,7]. Colonoscopy studies provide direct visualization of the colon and are commonly used for diagnostic and therapeutic purposes. While they offer real-time assessments of colon length, they may be subject to measurement errors and variations in procedural techniques. Magnetic endoscopic imaging (MEI) studies offer novel approaches to colonoscopy, allowing for enhanced visualization and measurement accuracy, particularly in challenging cases such as acromegaly. However, they may require specialized equipment and expertise, limiting their widespread applicability.

The meta-analysis performed here provided novel insights on the length of the colon and of its segments. The key messages were not only the actual mean values but also the large variability in lengths, with coefficients of variation between 19% and 39%. The greatest variability was seen in the sigmoid colon, most likely due to the fact that it is the most tortuous and is more difficult to image and assess. It is important to note that some of the variability in the measurement is also possibly originating from the different techniques and confounds mentioned above. Future research using state-of-the-art non-invasive techniques will be able to refine such data and is likely to reduce this variability.

Several studies indicated that colon length in adult males was longer than that in adult females, ranging between 4% and 18% longer [10,11,20,35] whilst others found no such differences [13,18] or the opposite difference [23,25], with the transverse colon being longer in females [15], which could be possibly due to case selection. Colon length also showed some functional association, with the colon in patients with constipation reported to be longer compared with people with normal bowel habits [2,27].

A better understanding of colon lengths and sex [25] or ethnic [34] differences may allow a better estimation of the likely ease of colonoscopy procedures, with increased lengths in the rectum, sigmoid, and total colon having been observed amongst patients with challenging incomplete procedures [12,14,15,21,24]. The demonstration of the varying length of the different segments supports previous impressions that the transverse colon is the longest segment and the most variable. This adds to our understanding of its key storage function. The large and variable size allows the accommodation of variable inputs to ensure regular bowel habits despite variable inputs being related to a differing intake of bulking agents like dietary fiber.

## 5. Conclusions

The data summarized here contribute to our understanding of colon morphology and may also have implications for clinical practice, particularly for colonoscopy and preoperative planning of surgical resections. It also has the potential to improve our understanding of anatomical correlates of functional diseases such as diarrhea and constipation. This work also highlighted the need for more standardized measurement protocols. There are much fewer studies on the colon in children and young adults and this should be a zone of focus for further research. Non-invasive, non-destructive imaging techniques such as MRI can provide more physiologically relevant measurements of colon length.

## Figures and Tables

**Figure 1 diagnostics-14-02190-f001:**
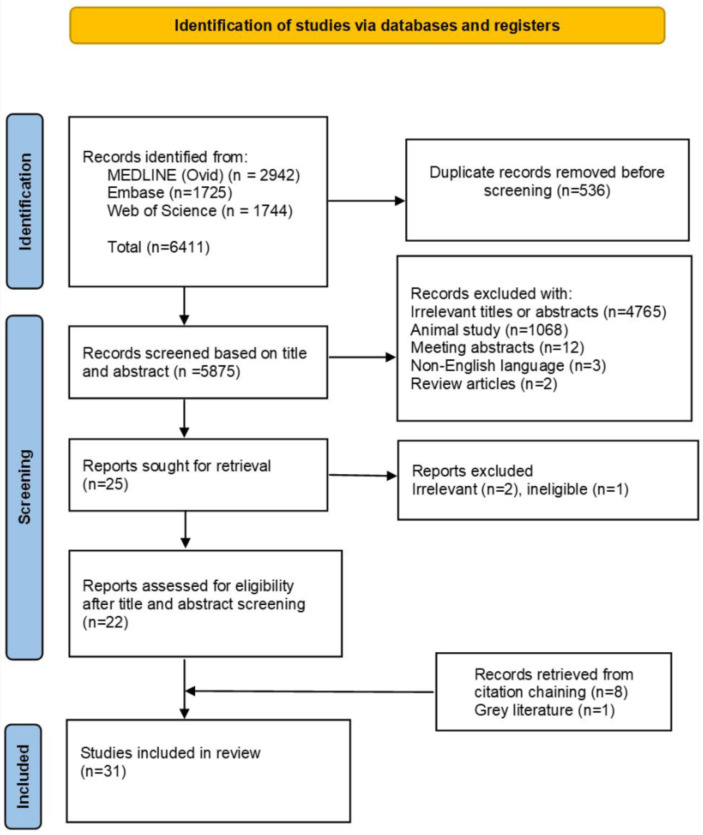
PRISMA flow diagram for systematic reviews showing the process used to identify the papers included in the systematic review.

**Figure 2 diagnostics-14-02190-f002:**
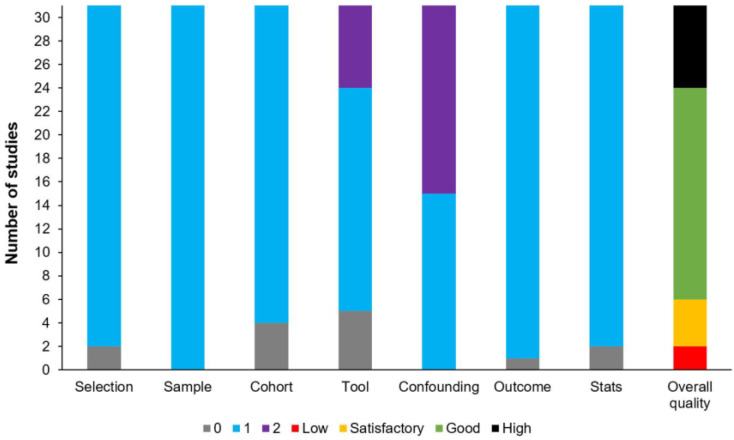
Newcastle–Ottawa Quality Assessment Scale for the studies included in the review.

**Figure 3 diagnostics-14-02190-f003:**
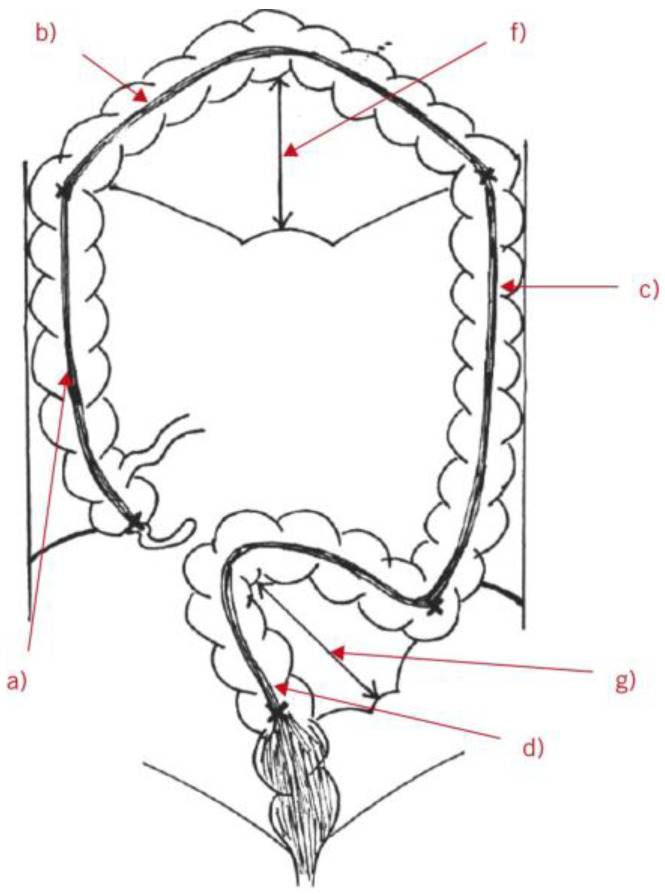
Diagram showing the colonic length measurements performed with a flexible tape measure on human cadavers that were fixed using a mixture of formaldehyde, ethanol, and methanol. The labels indicate, respectively, the following: (a) length of ascending colon (plus cecum); (b) length of transverse colon; (c) length of descending colon; (d) length of rectosigmoid colon; (f) height of transverse mesentery; (g) height of sigmoid mesentery. Reproduced with permission from Phiilips, Annals of the Royal College of Surgeons of England [18]; published by the Royal College of Surgeons of England, 2015.

**Figure 4 diagnostics-14-02190-f004:**
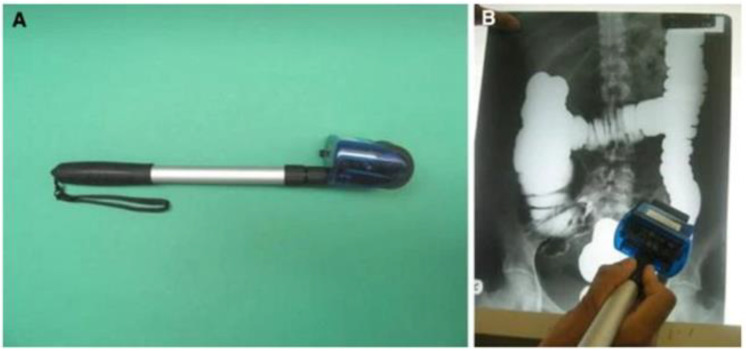
(**A**) Image of an opisometer. (**B**) Shows an investigator using an opisometer to measure the colon. The wheel measures length as it is run along the colon in the barium enema radiograph. Panel figure reproduced with permission from Madiba, Surgical and Radiologic Anatomy [34]; published by Wiley, 2008.

**Figure 5 diagnostics-14-02190-f005:**
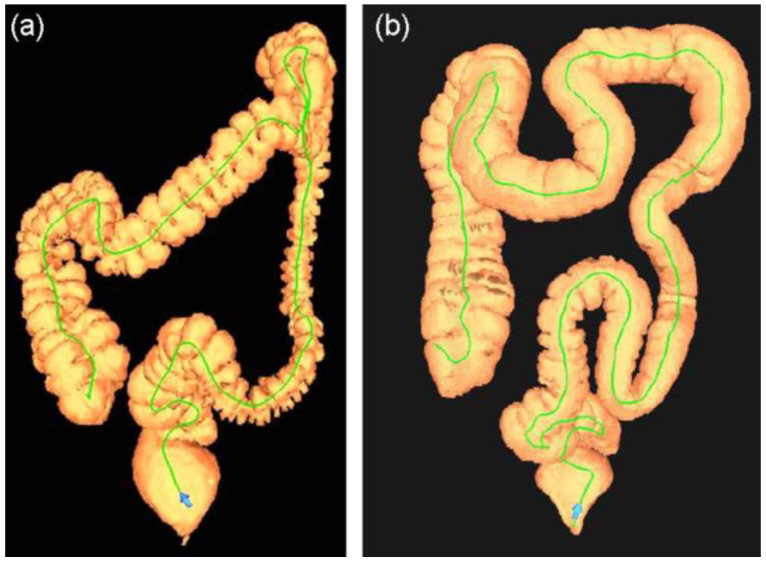
Examples of computed tomography colonography (CT) three-dimensional reconstructions of the colon. The green line shows the centerline calculated automatically for endoluminal navigation and colon length measurement. (**a**,**b**) illustrate different colon morphology and tortuosity in these two patients. Reproduced with permission from Eickhoff, Digestive and Liver Disease [24]; Wiley, 2010.

**Figure 6 diagnostics-14-02190-f006:**
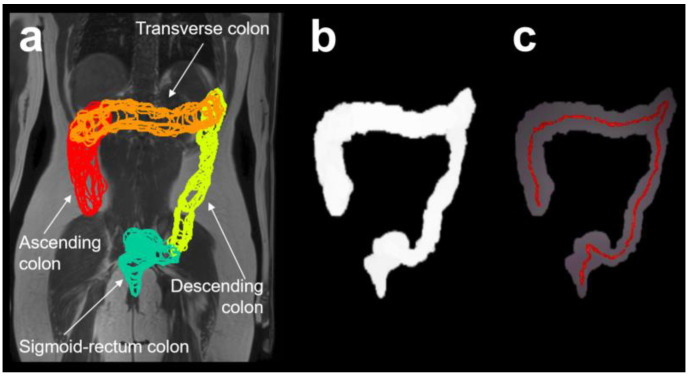
Example of MRI segmentation and 3D skeletonization measure of colon length. (**a**) A 2D representation of the regions of the colon superimposed to a coronal MRI scan of the body (ascending colon represented in red, transverse colon in orange, descending colon in yellow, and sigmoid/rectum colon in green). (**b**) A 2D projection of the overall binary mask for the entire colon. (**c**) A 2D projection of the 3D colon length path line (shown in red). Reproduced under Creative Commons Attribution License from reference [2].

**Figure 7 diagnostics-14-02190-f007:**
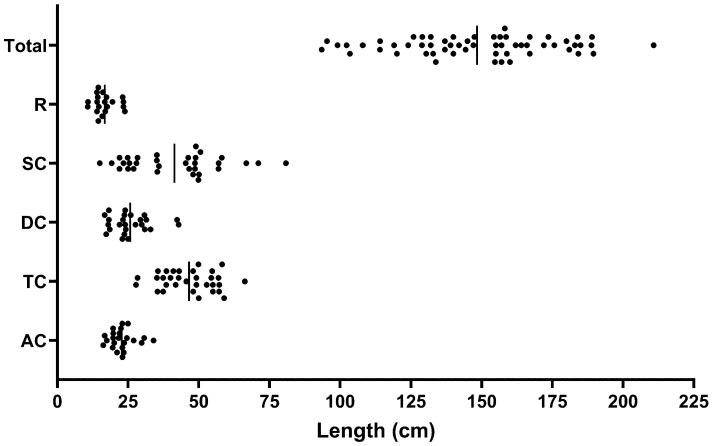
Mean values of the adult length of the colon and/or of its anatomical segments extracted from the reviewed literature. The solid vertical lines represent the overall mean value.

**Figure 8 diagnostics-14-02190-f008:**
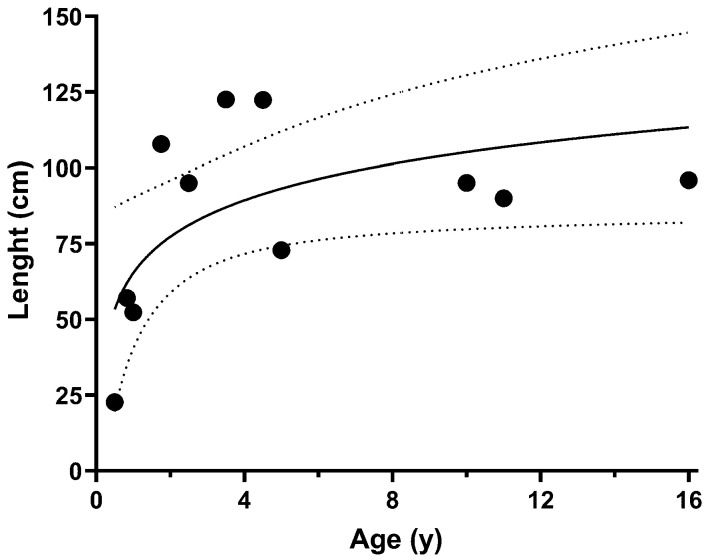
Mean pediatric total colon length of the colon extracted from the reviewed literature plotted against the corresponding mean age. The solid line indicates the best semi-logarithmic fit (R2 = 0.4012, *p* < 0.0333) and the dotted lines the 95% confidence interval.

**Table 1 diagnostics-14-02190-t001:** Overview of the included studies detailing population, demographics, and colon length measurement techniques employed.

Author	Year	Location	Study Design	Population	Control or Comparison	Sex	Age, y	Technique (Measurement Tool)
Treves, et al. [1]	1885	UK	Prospective	100 adult cadavers	NA	NA	NA	Dissection (unknown)
Bryant, et al. [2]	1924	USA	Prospective	45 fetal cadavers 37 child cadavers 160 adult cadavers	Comparison among age group and sex	Fetal: M: 25, F: 20 Children: M: 20, F: 17 Adult: M: 94, F: 66	Children (0.5–17) Adults (20–80)	Dissection (ruler)
Underhill, et al. [3]	1955	UK	Prospective	100 adult cadavers	NA	M: 65, F: 35	M: 27–91 F: 33–85	Dissection (ruler)
Chapuis, et al. [4]	1982	Australia	Retrospective	50 adult patients 10 cadavers	10 cadavers	Patients: M: 19, F: 31	Both sexes: 44	Barium enema (ruler); dissection (unknown)
Sadahiro, et al. [5]	1991	Japan	Retrospective	834 patients who underwent barium enema	NA	NA	NA	Barium enema (map measure)
Sadahiro, et al. [6]	1992	Japan	Retrospective	920 adult patients who underwent barium enema	Comparison between age, sex, and physique	M: 434 F: 486	M: 57.6 ± 14.3 (17–86) F: 56.1 ± 14.7 (14–92)	Barium enema (map measure)
Saunders, et al. [7]	1995	UK	Retrospective	48 adult patients who had a difficult colonoscopy and a barium enema	46 controls with no difficulty of colonoscopy and have a barium enema	Difficult colonoscopy: M: 15, F: 33 No difficult colonoscopy: M: 23, F: 23	Difficult colonoscopy: 64 (17–76) Patients with no difficult colonoscopy: 62 (23–81)	Barium enema (opisometer mapping wheel)
Saunders, et al. [8]	1995	UK	Prospective	118 adult laparotomy patients	NA	M: 66, F: 52	Both sexes: 63 (19–85)	Surgery (unknown)
Saunders, et al. [9]	1996	UK	Retrospective	345 adult patients	Comparison between M and F	M: 162, F: 183	M: 51.5 (15–85) F: 50.2 (19–85)	Barium enema (opisometer mapping wheel)
Rowland, et al. [10]	1999	UK	Retrospective	Colonoscopy Group 1: 156 patients no stiffening over tube aid; Group 2: 77 patients stiffening over tube aid	Comparison between M and F	Group 1: M: 76, F: 80 Group 2: M: 40, F: 37	M: 56.7 ± 15.1 F: 59.4 ± 13.0	Colonoscopy (computing system)
Hounnou, et al. [11]	2002	France	Prospective	200 adult cadavers	Comparison between age, weight, and height	M: 100, F: 100	Both sexes: 76 ± 12 M: 74 ± 12, F: 78 ± 12	Dissection (tape)
Bhatnagar, et al. [12]	2004	India	Prospective	51 surgical patients, 19 cadavers	Comparison between patients and cadavers	Patients: M: 27, F: 24 Cadavers: M: 17, F: 2	Patients: 36.16 ± 12.59 (16–60) Cadavers unknown	Surgery (tape)
Renehan, et al. [13]	2005	UK	Prospective	25 adult patients with acromegaly 41 adult controls	Non-acromegalic	Acromegaly: M: 56% F: 44% Control non-acromegalic: M: 60% F: 40%	Patients: 56 Control: 60	Colonoscopy (magnetic endoscopic imaging, MEI)
Hanson, et al. [14]	2007	US	Cross-sectional study	Adult patients CTC, 100 after incomplete colonoscopy, 100 after complete colonoscopy	Patients after complete optical colonoscopy	Incomplete colonoscopy: M: 41, F: 59 Complete colonoscopy: M: 60, F: 40	Incomplete colonoscopy: 63.4 ± 10.6 Complete optical colonoscopy: 58.2 ± 7.9	CTC (3D map with an automated centerline)
Madiba, et al. [15]	2008	South Africa	Retrospective	109 adult patients	Comparison between races	African: M: 16 F: 23 Indian: M: 25 F: 24 White: M: 7 F: 14	African: 52 (14–92) Indian: 48 (14–83) White: 61 (20–92)	Barium enema (opisometer mapping wheel)
Duncan, et al. [16]	2009	USA	Retrospective	338 adult patients who undergo CTC and OC	Comparison between different tools (OC and CT)	The majority are men (M-F ratio 1.8:1)	Both sexes: 58 (41–75)	CTC (automated centerline measurement and optical colonoscopy)
Khashab, et al. [17]	2009	USA	Retrospective	505 adult CTC patients	Comparison between age, sex, and body mass	Adults: M: 239, F: 266	Both sexes: 56.6 ± 7.3	CTC (automated centerline)
Punwani, et al. [18]	2009	UK	Retrospective	20 CTC patients with good colonic distention	Comparison between different imaging position	Adults: M: 10, F: 10	Both sexes: 54.6 ± 5.9	CTC (automated centerline)
Strujis, et al. [19]	2009	Canada	Prospective	108 laparotomy children	Comparison between age, height, and weight	NA	Children from 24 weeks up to 5 years	Surgery (silk suture)
Eickhoff, et al. [20]	2010	USA Germany	Retrospective CTC, prospective OC	Part 1: 100 adult CTC patients Part 2: 100 adult OC patients	Comparison between CT and OC	Part 1: M: 60, F: 40 Part 2: M: 57, F: 43	Part 1: 58.2 ± 7.9 Part 2: 60.4 ± 8.2	CTC (automated centerline measurement) and OC
Alatise, et al. [21]	2013	Nigeria	Prospective	50 adult patients 50 adult cadavers	Comparison between patients and cadavers	Patients: M: 25, F: 25 Cadavers: M: 25, F: 25	Living: 48.3 ± 1.7 Cadavers: 47.0 ± 2.0	Surgery (suture)
Michael, et al. [22]	2015	India	Prospective	31 adult cadavers	Comparison between M and F	M: 62%, F: 38%	(45–93)	Dissection (unknown)
Phillips, et al. [23]	2015	UK	Prospective	35 adult cadavers	Comparison between M and F	Adults: M:18, F:17	84 ± 13.2	Dissection (tape)
Alazmani, et al. [24]	2016	UK	Retrospective	24 adult patients	Different imaging position	Adults: M:12, F:12	54.8 ± 4.7 (50–56)	CTC (3D, automated centerline)
Ohgo, et al. [25]	2016	Japan	Retrospective	IBS-C: 13 IBS-D: 12 FC: 12 Control: 14	Healthy controls	Control: M: 6 F: 8 IBS-C: M: 6 F: 7 IBS-D: M: 10 F: 2 FC: M: 7 F: 5	Control: 64 IBS-D: 60 IBS-C: 61 FC: 70	CTC and OC (unknown)
Mark, et al. [26]	2017	Denmark	Prospective	Group 1: 25 healthy adults Group 2: 21 healthy adults	Comparison between different methods	Group 1: M: 25 Group 2: M: 10, F: 11	Group 1: 24 (21–56) Group 2: 38 (25–52)	3D-Transit (electromagnetic capsule tracking) MRI (semiautomated centerline)
Mirjalili, et al. [27]	2017	New Zealand	Retrospective	112 children	Comparison between different age groups	M: 51% F: 49%	33: <2 (1–23 months) 40: 4–6 years (49–65 months) 39: 9–11 years (110–127 months)	CTC (automated centerline)
Flor, et al. [28]	2020	Italy	Retrospective	144 adult patients without diverticula 323 adult patients with diverticula	Patients without diverticula	M: 177, F: 290	67 ± 12 (45–96)	CTC (interactive 3D map with an automated centerline)
Bayeh, et al. [29]	2021	Ethiopia	Prospective	Sigmoid volvulus patients. Group 1: 22 controls; Group 2: 22 elective surgery; Group 3: 22 emergency surgery	Patients who underwent surgery with no history of sigmoid volvulus	M: 56, F: 10	Group 1: 47.27 Group 2: 55.95 Group 3: 52.23	Surgery (tape)
Utano, et al. [30]	2022	Japan	Retrospective	295 adult patients:	Patients with positive fecal immunochemical tests	M: 154, F: 141	58.0 ± 11.0 (40–80)	CTC (automated centerline)
Sharif, et al. [31]	2024	UK	Retrospective	19 healthy children 16 patients with FC	Comparison with healthy controls	Healthy volunteers: M: 8, F: 11 FC: M: 7, F: 9	Controls: 16 ± 2 FC: 11 ± 3	MRI (3D skeletonization)

The data are presented as mean ± SD or median (range). CTC, computed tomography colonoscopy; F, female; FC, functional constipation; IBS, irritable bowel syndrome; M. male; NA, not avaialble; OC, optical colonoscopy.

**Table 2 diagnostics-14-02190-t002:** Measured values of the length of anatomical colon segment and/or total colon length listed by reviewed study with the corresponding colon length measurement techniques employed. Values are expressed in cm.

Study	CC	AC	TC	DC	SC	RC	Total Colon	Technique (Measurement Tool)
Treves, et al. [1]	NA	NA	NA	NA	NA	NA	M: 142 (142–198) F: 137 (99.06–198)	Dissection (unknown)
Bryant, et al. [2]	NA	NA	NA	NA	NA	NA	M: 162 (111–279) F: 137 (101–203)	Dissection (ruler)
Underhill, et al. [3]	NA	NA	NA	NA	NA	NA	M: 180 (140–198) F: 157 (140–182)	Dissection (ruler)
Chapuis, et al. [4]	Barium enema: 5.3 Postmortem: 7.5 M: 5.0, F: 5.6	Barium enema: 17.2 Postmortem: 17.1 M: 16.1, F: 17.9	Barium enema: 42.9 Postmortem: 54.8 M: 37.4, F: 46.3	Barium enema: 29.9 Postmortem: 31.5 M: 30.9, F: 29.4	Barium enema: 27 Postmortem: 23.4 M: 24.9, F: 28.4	Barium enema: 16.5 Postmortem: 23.9 M: 16.1 F: 16.8	Barium enema: 139 Postmortem: 158.2 M: 130.4 F: 144.3	Barium enema (ruler); dissection (unknown)
Sadahiro, et al. [5]	4.1 ± 0.8	15.7 ± 3.3	38.6 ± 8.5	18.2 ± 4.4	35.2 ± 10.1	17.4 ± 1.6	129 ± 15.8	Barium enema (map measure)
Sadahiro, et al. [6]	M: 4.04 ± 0.89 F: 4.19 ± 0.74	M: 15.50 ± 3.31 F: 15.93 ± 3.26	M: 35.44 ± 7.32 F: 41.87 ± 8.48	M: 18.56 ± 4.18 F: 18.00 ± 4.72	M: 35.37 ± 10.31 F: 35.22 ± 9.79	M: 16.96 ± 1.59 F: 17.62 ± 1.69	M: 125.87 ± 15.4 F: 132.83 ± 15.69	Barium enema (map measure)
Saunders, et al. [7]	NA	NA	NA	NA	NA	SC + RC: Patients: 61 Control: 53	Patients: 157 Control: 140	Barium enema (opisometer mapping wheel)
Saunders, et al. [8]	NA	CC + AC: 16.7 (7–30)	45.6 (25–81)	18.2 (9–42)	NA	SC + RC: 34 (17–78)	114.1 (68–159)	Surgery (unknown)
Saunders, et al. [9]	NA	AC + CC: M: 23 (15–38), F: 23 (11–41)	M: 40 (20–67) F: 48 (19–83)	M: 25 (8–36), F: 23 (11–43)	NA	SC + RC: M: 59 (31–103), F: 23 (22–100)	M: 145 (97–205) F: 155 (108–206)	Barium enema (opisometer mapping wheel)
Rowland, et al. [10]	NA	NA	M: 38.5 (11.1) F: 41.0 (13.6)	M: 23.7 (4.0) F: 23.2 (5.2)	NA	M: 14.7 (2.0) F: 14.5 (1.5)	NA	Colonoscopy (computing system)
Hounnou, et al. [11]	NA	NA	NA	NA	NA	NA	M: 166 (80–313) F: 155 (80–214)	Dissection (tape)
Bhatnagar, et al. [12]		NA	NA	NA	Patients: 46.6 ± 11.2 (25–86) Cadavers: 28 ± 7.6 (18.5–43)	NA	NA	Surgery (tape)
Renehan, et al. [13]	NA	NA	NA	NA	Patients: 25.5 (3.2) HC: 22.0 (2.2)	Patients: 15.9 (1.1) HC: 14.5 (1.2)	Patients: 132.9 (22.1) HC: 114.4 (16.1)	Colonoscopy (magnetic endoscopic imaging, MEI)
Hanson, et al. [14]	NA	NA	Incomplete colonoscopy: 66.3 ± 18.6 Complete colonoscopy: 49.2 ± 10.6	NA	Incomplete colonoscopy: 66.8 ± 22.2 Complete colonoscopy: 48.7 ± 13.6	NA	Incomplete colonoscopy: 210.8 ± 38.2 Complete colonoscopy: 167.0 ± 20.8	CTC (3D map with an automated centerline)
Madiba, et al. [15]	NA	NA	NA	NA	NA	SC + RC: African: M: 74 (25–88) F: 55 (44–73) Indian: M: 42 (25–65) F: 42 (22–67) White: M: 40 (24–71) F: 45 (24–62)	African: M: 160 (101–195) F: 140 (109–227) Indian: M: 120 (88–175) F: 124 (97–262) White: M: 119 (88–145) F: 132 (92–152)	Barium enema (opisometer mapping wheel)
Duncan, et al. [16]	NA	NA	NA	NA	NA	NA	CTC: 189 (75–257) OC: 108 (65–150)	CTC (automated centerline measurement and optical colonoscopy)
Khashab, et al. [17]	6.7 ± 1.9 (2–14)	23.1 ± 6.8 (9–62)	58.3 ± 13.6 (26–103)	33.0 ± 8.0 (18–75)	49.0 ± 12.9 (18–91)	19.5 ± 3.1 (7–28)	Both sexes: 189.5 ± 26.3 (120–299) M: 185.4 ± 26.5 (120–286) F: 193.3 ± 25.6 (135–299)	CTC (automated centerline)
Punwani, et al. [18]	NA	Prone: M: 30.72 ± 6.67 F: 23.07 ± 5.43 Supine: M: 34.01 ± 9.08 F: 23.68 ± 9.65	Prone: M: 49.99 ± 5.32 F: 52.86 ± 14.23 Supine: M: 48.08 ± 4.51 F: 54.44 ± 12.76	Prone: M: 42.93 ± 13.31 F: 30.8 ± 9.21 Supine: M: 42.34 ± 13.07 F: 27.68 ± 8.94	Prone: M: 46.28 ± 16.08 F: 58.18 ± 11.92 Supine: M: 45.43 ± 13.22 F: 57.08 ± 9.49	Prone: M: 14.30 ± 3.10 F: 10.83 ± 2.55 Supine: M: 14.09 ± 2.95 F: 10.73 ± 3.21	Prone: M: 184.2 ± 26.9 F: 175.7 ± 33.3 Supine: M: 183.9 ± 22.6 F: 173.6 ± 32.7	CTC (automated centerline)
Strujis, et al. [19]	NA	NA	NA	NA	NA	NA	24–26 wk: 22.7 ± 2.0 27–29 wk: 24.4 ± 1.2 30–32 wk: 37.7 ± 2.2 33–35 wk: 27.8 ± 1.7 36–38 wk: 40.1 ± 4.3 39–40 wk: 32.7 ± 2.1 0–6 mo: 56.8 ± 2.7 7–12 mo: 57.1 ± 2.2 13–18 mo: 84.8 ± 2.3 19–24 mo: 107.8 ± 4.5 25–36 mo: 95.0 ± 3.4 37–48 mo: 122.5 ± 5.9 49–60 mo: 122.4 ± 5.7	Surgery (silk suture)
Eickhoff, et al. [20]	NA	NA	49.2 ± 10.6	NA	48.7 ± 13.6	NA	CTC: 167.0 ± 20.8 OC: 93.5 ± 15.3	CTC (automated centerline measurement) and OC
Alatise, et al. [21]		NA	NA	NA	Patients: 48.9 Cadavers: 50.1	NA	NA	Surgery (suture)
Michael, et al. [22]	NA	NA	NA	NA	Mesenteric border pelvis brim: 15 ± 4.45 Mesentery root: 19.2 ± 6 Antimesenteric border pelvis brim: 22 ± 7.9 Mesentery root: 25 ± 8.7.	NA	NA	Dissection (unknown)
Phillips, et al. [23]	NA	CC + AC: 20.9 ± 4.7	50.2 ± 9.5	21.8 ± 5.4	NA	SC + RC: 38.3 ± 10.5	M: 133.8 ± 3.7 F: 128.9 ± 2.7	Dissection (tape)
Alazmani, et al. [24]		Supine: 21.7 ± 4.2 (20.7) Prone: 19.7 ± 4.0 (20.3)	Supine: 57.2 ± 9.3 (56.6) Prone: 57.3 ± 10.9 (56.9)	Supine: 24.2 ± 7.8 (23.1) Prone: 26.0 ± 7.8 (25.9)	Supine: 50.6 ± 13.9 (51.6) Prone: 49.9 ± 11.7 (48.7)	Supine: 23.4 ± 6.7 (21.7) Prone: 23.1 ± 3.9 (22.7)	Supine: 185.0 ± 18.3 (187.5) Prone: 183.0 ± 16.9 (185.0)	CTC (3D, automated centerline)
Ohgo, et al. [25]	NA	NA	IBS-D: 43.1 IBS-C: 57.0 FC: 55.0 C: 49.9	NA	NA	SC + RC: IBS-D: 55.9 IBS-C: 63.6 FC: 77.4 C: 56.2	IBS-D: 158.9 IBS-C: 172.0 FC: 188.8 C: 156.5	CTC and OC (unknown)
Mark, et al. [26]	NA	CC + AC: MRI: 16.2 ± 3.5 3D-Transit: 22.0 ± 7.5 3D-Transit day 1: 17.5 ± 6.7 3D-Transit day 2: 19.7 ± 6.7	MRI: 27.8 ± 5.4 3D-Transit: 28.4 ± 4.7 3D-Transit day 1: 35.2 ± 5.4 3D-Transit day 2: 35.6 ± 6.5	MRI: 23.7 ± 4.1 3D-Transit: 24.0 ± 7.4 3D-Transit day 1: 17.3 ± 3.4 3D-Transit day 2: 16.7 ± 4.9	NA	SC + RC: MRI: 27.8 ± 11.2 3D-Transit: 24.7 ± 8.7 3D-Transit day 1: 32.3 ± 10.8 3D-Transit day 2: 31.4 ± 9.2	MRI: 95.4 ± 14.6 3D-Transit: 99.1 ± 17.9 3D-Transit day 1: 102.3 ± 13.3 3D-Transit day 2: 103.5 ± 15.1	3D-Transit (electromagnetic capsule tracking) MRI (semiautomated centerline)
Mirjalili, et al. [27]	NA	CC + AC: 0–2 years: 7.4 ± 3.4 4–6 years: 12.1 ± 3.4 9–11 years: 13.5 ± 2.9	0–2 years: 16.4 ± 3.0 4–6 years: 19.8 ± 4.7 9–11 years: 28.0 ± 7.7	0–2 years: 9.6 ± 3.6 4–6 years: 14.8 ± 4.5 9–11 years: 21.2 ± 5.0	0–2 years: 14.6 ± 6.2 4–6 years: 17.7 ± 7.6 9–11 years: 22.3 ± 7.5	0–2 years: 4.4 ± 1.4 4–6 years: 8.7 ± 2.2 9–11 years: 10 ± 2.3	0–2 years: 52.3 ± 10.9 4–6 years: 72.9 ± 11.4 9–11 years: 95.1 ± 12.6	CTC (automated centerline)
Flor, et al. [28]	Overall: 4 ± 1 Patients: 4 ± 1 HC: 5 ± 1	Overall: 21 ± 5 Patients: 21 ± 5 HC: 22 ± 6	Overall: 56 ± 13 Patients: 55 ± 12 HC: 59 ± 14	Overall: 22 ± 7 Patients: 22 ± 6 HC: 24 ± 8	Overall: 51 ± 14 Patients: 48 ± 13 HC: 57 ± 16	Overall: 14 ± 2 Patients: 14 ± 2 HC: 14 ± 2	Overall: 169 ± 25 Patients: 164 ± 22 HC: 181 ± 27	CTC (interactive 3D map with an automated centerline)
Bayeh, et al. [29]		NA	NA	NA	No history of volvulus: 35.91 Non-surgical detorsion of volvulus: 71.07 Emergency surgery for sigmoid volvulus: 80.86	NA	NA	Surgery (tape)
Utano, et al. [30]	NA	NA	NA	NA	NA	NA	Daily defecation 147.4 ± 17.9 Defecation every 2–3 days: 154.7 ± 18.5 Defecation < than/3 days: 158.6 ± 18.3 F:1 54.3 ± 18.1 M: 147.1 ± 18.3 Average: 150.3 ± 18.5	CTC (automated centerline)
Sharif, et al. [31]	NA	CC + AC: HC: 19 ± 1 FC: 17 ± 1	HC: 27 ± 1 FC: 24 ± 2	HC: 31 ± 2 FC: 26 ± 1	SC + RC: HC: 20 ± 1 FC: 22 ± 2	NA	HC: 96 ± 3 FC: 90 ± 5	MRI (3D skeletonization)

All length values are expressed in cm as mean ± SD or as median (range). AC, ascending colon; CC, cecum colon; CTC, computed tomography colonography; DC, descending colon; F, female; HC, healthy control; M, male; mo, month; NA, not available; OC, optical colonoscopy; RC, rectum colon; RSC, rectosigmoid colon; SC, sigmoid colon, TC: transverse colon; wk, week.

**Table 3 diagnostics-14-02190-t003:** Descriptive summary values of the length of the colon and/or of its anatomical segments extracted from the reviewed literature.

Values	AC	TC	DC	SC	R	Total
Number of values	23	32	25	32	20	58
Minimum	16.2	27.8	16.7	15.0	10.7	93.5
25% Percentile	19.7	38.5	20.3	25.9	14.1	130.1
Median	22.5	48.0	24.0	45.9	16.0	150.9
75% Percentile	24.6	55.0	30.3	50.1	19.0	167.0
Maximum	34.0	66.3	42.9	80.9	23.9	210.8
Range	17.8	38.5	26.2	65.9	13.2	117.3
Mean	22.8	46.6	25.8	41.4	16.8	148.3
Std. Deviation	4.4	9.6	7.0	16.2	4.0	27.1
Std. Error of mean	0.9	1.7	1.4	2.9	0.9	3.8
Coefficient of variation	19.3%	20.6%	27.1%	39.2%	23.8%	18.3%

## Data Availability

No new data were created or analyzed in this study. Data sharing is not applicable to this article.

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
