# Peer review of "Experimental Measurements of the Length of the Human Colon: A Systematic Review and Meta-Analysis"

_diagnostics, 2024, doi:10.3390/diagnostics14192190_

Round 1

Reviewer 1 Report

Comments and Suggestions for Authors

1.This paper analyzes data from 31 studies, providing valuable insights into the current understanding of the length of the human colon and its anatomical segments. It serves as a useful adjunct for preoperative planning in procedures such as colonoscopy and surgical resection.

2.In addition to anatomical length, the length of the colon may also involve functional changes influenced by neurological factors. Therefore, discrepancies may exist between measurements taken from living individuals and cadavers, and even among living individuals depending on whether invasive or non-invasive methods are employed. These differences cannot be overlooked.

3.It is recommended that the abstract be structured logically, following the format of purpose, methods, results, and conclusions.

4.The included studies span from 1885 to 2024, during which time significant advancements have occurred in medical technology and measurement methods. It is questionable whether the studies from earlier years are comparable to those conducted today. Additionally, differences among populations from various countries have not been explicitly considered. These aspects should be given due consideration.

Comments on the Quality of English Language

 Minor editing of English language required.

Author Response

Diagnostics: Manuscript ID: diagnostics-3207036

Experimental measurements of the length of the human colon: a systematic review and meta-analysis

REVIEWER 1

We are grateful to this Reviewer for their constructive comments, which we have taken on board and answered as detailed below and highlighted in yellow on the revised manuscript. We hope that with the changes made this Revier will find our manuscript suitable for publications in Diagnostics.

Comments and Suggestions for Authors

1.This paper analyzes data from 31 studies, providing valuable insights into the current understanding of the length of the human colon and its anatomical segments. It serves as a useful adjunct for preoperative planning in procedures such as colonoscopy and surgical resection.

Answer: we are grateful for this positive appraisal of our manuscript.

2.In addition to anatomical length, the length of the colon may also involve functional changes influenced by neurological factors. Therefore, discrepancies may exist between measurements taken from living individuals and cadavers, and even among living individuals depending on whether invasive or non-invasive methods are employed. These differences cannot be overlooked.

Answer: We agree that this is an important point and that it could have been made clearer. In response to this point we have rewritten more strongly parts of the discussion, namely Lines 356-359 and Lines 389-392.

3.It is recommended that the abstract be structured logically, following the format of purpose, methods, results, and conclusions.

Answer: Thank you for this suggestion, we have now structured the abstract as recommended.

4.The included studies span from 1885 to 2024, during which time significant advancements have occurred in medical technology and measurement methods. It is questionable whether the studies from earlier years are comparable to those conducted today. Additionally, differences among populations from various countries have not been explicitly considered. These aspects should be given due consideration.

Answer: As above, we agree with this point and have answered accordingly at point number 2 of this review.  

Comments on the Quality of English Language

Minor editing of English language required.

Answer: Thank you, we have revised some minor aspects of the language and spelt checked the manuscript again.

Reviewer 2 Report

Comments and Suggestions for Authors

Knowledge of the length of the colon contributes to an understanding of its morphology and function and has implications for clinical practice. This article reviewed various methods of measuring colon length and their results, and compared the characteristics of different methods, ultimately concluding that more standardized measurement protocols and more child-specific studies are needed in the future. Overall the article is well organized and has clinical relevance, but there are still the following issues that could be further revised.

Introduction

The description of the anatomy of the colon, such as anatomic landmarks for the beginning and end of the colon and its segments, could probably be added.

Results

It is recommended that the methods of measurement be labeled in Table 2, i.e., whether it is measured directly during surgical procedures and on cadavers or indirectly using colonoscopy and radiological techniques.

It is suggested that the order of 3.4.4 Intraoperative assessment and 3.4.2 Radiological barium enema assessment studies be reversed, with postmortem and intraoperative measurements being direct measurements and from 3.4.4 onwards being radiological techniques.

As in the 3.5 Meta-analysis Section, there was a large variability in lengths among studies, has any consideration been given to the source of these differences and the implications for future research and clinical applications?

Author Response

Diagnostics: Manuscript ID: diagnostics-3207036

Experimental measurements of the length of the human colon: a systematic review and meta-analysis

REVIEWER 2

We are grateful to this Reviewer for their constructive comments, which we have taken on board and answered as detailed below and highlighted in yellow on the revised manuscript. We hope that with the changes made this Reviewr will find our manuscript suitable for publications in Diagnostics.

Comments and Suggestions for Authors

Knowledge of the length of the colon contributes to an understanding of its morphology and function and has implications for clinical practice. This article reviewed various methods of measuring colon length and their results, and compared the characteristics of different methods, ultimately concluding that more standardized measurement protocols and more child-specific studies are needed in the future. Overall the article is well organized and has clinical relevance, but there are still the following issues that could be further revised.

Answer: Thank you so much for the positive appraisal of our work, we have addressed the issues raised as detailed below.

Introduction

The description of the anatomy of the colon, such as anatomic landmarks for the beginning and end of the colon and its segments, could probably be added.

Answer: Thank you for this comments, agreed, we have added an anatomical description of the colon and of the segmental landmarks in Introduction, Lines 48-54.

Results

It is recommended that the methods of measurement be labeled in Table 2, i.e., whether it is measured directly during surgical procedures and on cadavers or indirectly using colonoscopy and radiological techniques.

Answer: Thank you for this suggestion, we agree and the measurement techniques have now been listed in Table 2 in the rightmost column.

It is suggested that the order of 3.4.4 Intraoperative assessment and 3.4.2 Radiological barium enema assessment studies be reversed, with postmortem and intraoperative measurements being direct measurements and from 3.4.4 onwards being radiological techniques.

Answer: Agreed, the two paragraphs have been swapped as suggested.

As in the 3.5 Meta-analysis Section, there was a large variability in lengths among studies, has any consideration been given to the source of these differences and the implications for future research and clinical applications?

Answer: In keeping also with Reviewer 1’s comments, we agree that this is an important point and that it could have been made clearer. In response to this point we have specifically rewritten more strongly parts of the discussion, namely Lines 356-359 and Lines 389-392.

Round 2

Reviewer 1 Report

Comments and Suggestions for Authors

Your thoroughness and responsiveness in addressing the feedback have significantly enhanced the quality and clarity of the work. Congratulations on your efforts, and I eagerly anticipate the publication of your valuable contribution.